# Combination of Mesenchymal Stem Cell-Delivered Oncolytic Virus with Prodrug Activation Increases Efficacy and Safety of Colorectal Cancer Therapy

**DOI:** 10.3390/biomedicines9050548

**Published:** 2021-05-13

**Authors:** Chun-Te Ho, Mei-Hsuan Wu, Ming-Jen Chen, Shih-Pei Lin, Yu-Ting Yen, Shih-Chieh Hung

**Affiliations:** 1Drug Development Center, Institute of New Drug Development, Institute of Biomedical Sciences, School of Medicine, China Medical University, Taichung 404, Taiwan; homenjer@gmail.com (C.-T.H.); d92449001@ntu.edu.tw (Y.-T.Y.); 2Integrative Stem Cell Center, Department of Orthopaedics, China Medical University Hospital, Taichung 404, Taiwan; 3Institute of Clinical Medicine, School of Medicine, National Yang-Ming University, Taipei 112, Taiwan; hsuan0425@hotmail.com (M.-H.W.); cutebettykimo@yahoo.com.tw (S.-P.L.); 4Department of Surgery, MacKay Memorial Hospital & Department of Medicine, MacKay Medical College, New Taipei City 252, Taiwan; mjchen@mmc.edu.tw

**Keywords:** oncolytic virus, mesenchymal stem cell, prodrug activation, p53 mutant tumor, colorectal cancer

## Abstract

Although oncolytic viruses are currently being evaluated for cancer treatment in clinical trials, systemic administration is hindered by many factors that prevent them from reaching the tumor cells. When administered systemically, mesenchymal stem cells (MSCs) target tumors, and therefore constitute good cell carriers for oncolytic viruses. MSCs were primed with trichostatin A under hypoxia, which upregulated the expression of CXCR4, a chemokine receptor involved in tumor tropism, and coxsackievirus and adenovirus receptor that plays an important role in adenoviral infection. After priming, MSCs were loaded with conditionally replicative adenovirus that exhibits limited proliferation in cells with a functional p53 pathway and encodes Escherichia coli nitroreductase (NTR) enzymes (CRAdNTR) for targeting tumor cells. Primed MSCs increased tumor tropism and susceptibility to adenoviral infection, and successfully protected CRAdNTR from neutralization by anti-adenovirus antibodies both in vitro and in vivo, and specifically targeted p53-deficient colorectal tumors when infused intravenously. Analyses of deproteinized tissues by UPLC-MS/QTOF revealed that these MSCs converted the co-administered prodrug CB1954 into cytotoxic metabolites, such as 4-hydroxylamine and 2-amine, inducing oncolysis and tumor growth inhibition without being toxic for the host vital organs. This study shows that the combination of oncolytic viruses delivered by MSCs with the activation of prodrugs is a new cancer treatment strategy that provides a new approach for the development of oncolytic viral therapy for various cancers.

## 1. Introduction

Colorectal cancer is a major global health problem owing to its high prevalence and mortality rates, and it is the fourth most common cancer worldwide, including in many parts of the western world [1]. The most prevalent problem in cancer therapy is the regrowth and metastasis of malignant cells after standard treatment with surgery, radiation, and/or chemotherapy. Colorectal cancer recurrence can occur either locally or at a distant site; therefore, metastatic colorectal cancer remains difficult to treat effectively.

Oncolytic virotherapy is an emerging treatment modality with great therapeutic potential that uses replication-selective viruses to replicate specifically in tumors to destroy tumor cells by cytolysis [2]. Once the tumor cells are destroyed, viral progenies are released, allowing them to continuously hijack the neighboring tumor cells. This property is either inherent or genetically engineered [3]. Human adenovirus serotype 5 (Ad5) is one of the most widely studied engineered oncolytic viruses in preclinical studies and clinical trials [4]. When adenoviruses target host cells, coxsackievirus and adenovirus receptor (CAR) plays an important role in the infection process [5,6]. Conditionally replicative adenovirus (CRAd) has recently been used in preclinical studies and clinical trials to treat a broad range of cancers [7]. ONYX-015, an adenovirus made conditionally replicative by E1B-55k gene deletion [8,9], is a potential candidate for treatment of p53-deficient cancers, with minimum replication in cells with a functional p53 pathway [10]. The safety and viral pharmacokinetics of CRAd for cancer therapy had been studied in early phase clinical trials [11] and the viral genomes in plasma were only detected in early hours following IV infusion with 2 × 10^11^ to 2 × 10^12^ particles and were not detected at 8 h [8]. However, its antitumor efficacy has not been well documented, since there are many systemic barriers that prevent it from reaching tumor cells when administered systemically [12].

Multipotent stromal cells (MSCs) possessing self-renewal and multipotent capacities are inherently tumor-homing [13,14,15], and can be isolated, expanded, and transduced, making them viable candidates for cell carriers to deliver oncolytic viruses [16]. The biodistribution and cell-fate tracking of human MSCs following IV infusion into mice bearing human tumor xenografts have been well studied before [13]. They were mainly detected at the tumor inoculation sites, grew as the tumors grew, and then differentiated into tumor stroma, endothelial cells of tumor capillaries and small blood vessels, hair follicle cells, and basal cells of dermis overlaying the s.c. tumor xenografts [13]. The migration of MSCs is induced by several chemokine/chemokine receptor-mediated signaling pathways, in which the chemokine receptor CXCR4 plays a crucial role with regard to tumor tropism, as it allows dynamically interacting with the host by responding to stromal cell-derived factor-1 (SDF-1) secreted by tumors [17]. Previous studies have also shown that MSCs expanded under hypoxic conditions (referred to as hypoxic MSCs) exhibit greater proliferation, reduced senescence, increased stemness, and increased homing to tumor sites via upregulation of CXCR4 [16,18,19]. Moreover, hypoxic MSCs are immunologically privileged, and could be allotransplanted, without causing severe adverse effects [20].

Gene-directed enzyme-prodrug therapy (GDEPT), also known as suicide gene therapy, is based on the delivery of a foreign gene that encodes a prodrug-activating enzyme, followed by systemic administration of a nontoxic prodrug that is subsequently converted into a potent cell-killing drug [21]. The concept of using a bacterial nitroreductase (NTR) enzyme to react with its substrate prodrug 5-(azaridin-1-yl)-2,4-dinitrobenzamide (CB1954), and produce potent DNA cross-linking intermediates that can induce apoptosis in both dividing and non-dividing cells, is a promising and emerging GDEPT strategy that has reached clinical trials [22,23]. According to this evidence, the combination of CRAd and CB1954 is a new cancer treatment strategy for p53-deficient cancers. However, it remains to be elucidated whether the use of MSC as a cell carrier can further improve its anti-cancer efficacy and safety. In the current study, we first improved the susceptibility of MSCs to CRAd infection, and further increased MSC tumor tropism. We then combined MSC-delivered CRAd and prodrug activation to kill p53-deficient colorectal cancer cells both in vitro and in vivo. Through this combinational approach, we are able to increase both efficacy and safety of cancer therapy.

## 2. Materials and Methods 

### 2.1. Cell Culture

Human primary MSCs from bone marrow were harvested with the approval of Institutional Review Board and prepared as described previously [24]. The cells were seeded at 50 cells/cm^2^ and grown in Dulbecco’s modified eagle medium (DMEM, Gibco, Paisley, Scotland, UK), supplemented with 10% fetal bovine serum (FBS; Gibco, Paisley, Scotland, UK), 100 units/mL penicillin G, 100 μg/mL streptomycin sulfate and 25 μg/mL amphotericin B (Antibiotic Antimycotic Solution, Sigma-Aldrich, St. Louis, MO, USA). For the hypoxic culture, MSCs were maintained as previously described [19]. The MSCs were cultured in a gas mixture composed of 94% N_2_, 5% CO_2_, and 1% O_2_, which was maintained using an incubator with two air sensors: one for CO_2_ and the other for O_2_. The O_2_ concentration was achieved and maintained via the delivery of N_2_ from a tank containing pure N_2_. If the percentage of O_2_ exceeded the desired level, N_2_ gas was automatically injected into the system, to displace the excess O_2_. The passage number of MSCs used in this study was no more than five. MSCs were treated with 100 ng/mL Ttichostatin A (TSA, Sigma-Aldrich, St. Louis, MO, USA) for 24 h given a substitution name as primed MSCs, applied in a subsequent study. HT29, SW480, and SW620, p53-mutant human colorectal cancer cell lines [25], were obtained from the American type culture collection (ATCC, Rockville, MD, USA) and grown in DMEM supplemented with 10% FBS. For proliferation assay, cells were seeded at a density of 1000 cells/well in 96-well plate, followed by recover and assay with WST-1 kit (Cayman Chemical Company, Ann Arbor, MI, USA) at indicated time periods. For osteogenesis, cells were induced in DMEM supplemented with 10% FBS, 10^−8^ M dexamethasone (D1756, Sigma-Aldrich, Milan, Italy), 50 ng/mL ascorbic acid (A4406, Sigma-Aldrich, Milan, Italy), and 10^−2^ M β-glycerophosphate (50020, Fluka, Buchs, Switzerland) for 14 days. For adipogenesis, cells were induced with DMEM supplemented with 10% FBS, 0.05 mM indomethacin (I7378, Sigma-Aldrich, Milan, Italy), 10 μg/mL insulin (I0516, Sigma-Aldrich, Milan, Italy), 10^−8^ M dexamethasone, 50 μg/mL ascorbic acid, and 0.45 mM 3-isobutyl-1-methylxanthine (IBMX, I5879, Sigma-Aldrich, Milan, Italy) for 14 days. For chondrogenesis, 2.5 × 10^5^ cells/mL was transferred into 15-mL tube (5540300, Orange Scientific, Braine-l’Alleud, Belgium) and spun down. After 24 h, the cell pellet will form ball-like micromass and transfered the medium into DMEM (SH30081, Hyclone, Logan, UT, USA) with 1 mM dexamethasone, 1% NEAA (M7145, Sigma-Aldrich, Milan, Italy), 0.1% ITS+ (I2521, Sigma-Aldrich, Milan, Italy), 50 μg/mL ascorbic acid, 40 μg/mL L-proline (P5607, Sigma-Aldrich, Milan, Italy), 100 μg/mL sodium pyruvate (P5280, Sigma-Aldrich, Milan, Italy), and 10 ng/mL TGF-β1 (T7039, Sigma-Aldrich, Milan, Italy) contained. The micromass was cultured by shaking every other day to prevent adherence to the tube and culture for 21 days. Osteogenic, adipogenic differentiation, and chondrogenic micromass sections were detected by staining with Alizarin red S (A5533, Sigma-Aldrich, Milan, Italy), Oil red O (O9755, Sigma-Aldrich, Milan, Italy), and Alcian blue (B8438, Sigma-Aldrich, Milan, Italy) / nuclear fast red (N3020, Sigma-Aldrich, Milan, Italy), respectively.

### 2.2. Flow Cytometry

To confirm the MSCs phenotype and to analyze the effect on the surface expression level of CXCR4 and CAR after TSA treatment, the MSCs were plated at 2 × 10^5^ per 10-cm dish recovered for 24 h. The MSCs were treated with 100 ng/mL TSA for 24 h. After 24 h, the primed MSCs were washed with phosphate-buffered saline (PBS) and were harvested with Accutase^TM^ (StemCell Technologies Inc., Vancouver, BC, Canada) and incubated with control rabbit IgG (eBioscience, San Diego, CA, USA), or polyclonal antibodies against human CXCR4 (1:50, Novus Biologicals, Littleton, CO, USA) and CX3CR1 (1:100, Abcam, Cambridge, UK), CAR (1:100, Merck KGaA, Darmstadt, Germany), CD29 (1:50, Ancell, Bayport, MN, USA), CD31 (1:50, Ancell, Bayport, MN, USA), CD34 (1:50, Ancell, Bayport, MN, USA), CD44 (1:50, BD PharMingen, San Diego, CA, USA), CD45 (1:20, eBioscience, San Diego, CA, USA), or CD105 (1:50, Ancell, Bayport, MN, USA), at 4 °C for 1 h. After the PBS wash, the cells were incubated with PE-conjugated, goat anti-rabbit IgG, at 4 °C for 30 min. The samples were then analyzed on a Becton Dickinson FACSCalibur flow cytometer and data analysis was performed using CellQuest Pro software v. 4.0.2 (BD Biosciences, Sydney, Australia).

### 2.3. In Vitro Migration Assay

HT29 human colorectal cancer cell line was plated onto 24-well plates at 1 × 10^5^ cells per well in DMEM supplemented with 10% FBS for 24 h before the migration assay was performed. Hypoxic MSCs were treated with 100 ng/mL TSA for 24 h, then MSCs and hypoxic primed MSCs were seeded onto the 8 μm pore-size cell culture inserts (Corning Inc., Corning, NY, USA), at 5000 cells per well, in DMEM supplemented with 10% FBS, and allowed to adhere for 1 h at 37 °C. The inserts were then added to the lower chambers with or without HT29 and incubated at 37 °C, with 5% CO_2_. After 14 h of migration, the medium was removed and cells that remained attached to the upper surface of the insert filters were removed, using cotton swabs. The cells that had migrated to the opposite side of the filters were stained with Fluoroshield™ with DAPI (Sigma-Aldrich, St. Louis, MO, USA) and counted under a fluorescent microscope at 100× magnification. The average number of migrating cells per field was assessed by counting six random fields per filter.

### 2.4. Adenovirus Infection

Hypoxic primed MSCs were trypsinized and counted the cell numbers, then infected with CRAdNTR (PS1217H6), kindly gifted by Dr. Ming-Jen Chen of Mackay Memorial Hospital, at a multiplication of infection (MOI) of 100 vp/cell, in serum free α-MEM, for 1.5 h at 37 °C, and then an equal volume of growth medium (10% FBS) was added. After 24 h of infection, the medium was replaced with fresh culture medium, containing 10% FBS. CRAdNTR loaded-MSCs were indicated to MSC^CRAdNTR^. For investigation of adenoviral infection rate, CRAdEGFP was used in the same method to MSCs.

### 2.5. Combination Cytotoxicity Assay

To determine the cytotoxic effect of CB1954 in combination with MSCs-delivered CRAdNTR, HT29 cells were plated in a 24-well plate at 2 × 10^6^ cells per well for 24 h. The hypoxic primed MSCs, MSC^CRAdNTR^ for 48 h in a ratio of 1:10 and 2 × 10^8^ CRAdNTR (MOI = 100) were direct co-cultured with HT29 cells in presence of 50 μl high titer human serum diluted in 2 mL DMEM per well of a 24-well plate. The mixed culture cells were incubated at 37 °C for 2 days. Then, cells were treated with 10 μM CB1954 for 2 days, before the determination of cell viability using the tetrazolium-based colorimetric assay (MTT) and crystal violet staining. At different times during the culture, the medium was removed, the cells were washed once with PBS and 900 μL of medium and 100 μL of MTT solution (5 mg/mL) was added into each well and incubated at 37 °C for 4 h. At the end of incubation, the medium was aspirated and 200 μL of DMSO was added to dissolve the dye. The absorbance at O.D.570 was measured using Tecan microplate reader (Infinite M1000, Tecan). The results were shown as the mean of three independent replicates ± SD. The control wells without cells were used to account for the colorant retained by the culture plates. For crystal violet staining, the cells were fixed with 10% buffered formalin for 5 min and stained with 1% crystal violet, at room temperature for 20 min. Images of the plates were then captured with a scanner. The MTT assay and the crystal violet were performed in parallel.

### 2.6. Genomic DNA Extraction

The excision of tissue and organs (20 mg) from a mouse including tumor, heart, liver, lung, and kidney were quickly minced and ground with micropestles. For adherent cells, the cells were trypsinized before harvesting. The cells were then centrifuged for 30 s at 13,000× *g*, to discard the supernatant. The EasyPure Genomic DNA mini kit (Bioman Scientific Co., New Taipei City, Taiwan) was used to extract genomic DNA. Aliquots of 600 μL cell lysis solution were added to the extracted sample. For tissue extraction, proteinase K (4 mg/mL, Sigma-Aldrich, St. Louis, MO, USA) was added to the lysate and mixed by inverting 25 times and incubated at 55 °C for 3 h to overnight, until tissue particles have dissolved. For lysing the cells, pipetting up and down was made until no visible cell clumps remain, then 3 μL of 20 mg/mL RNase A was added to the nuclear lysate and the sample was mixed by inverting the tubes 25 times and incubated for 15–30 min at 37 °C, followed by adding 200 μL of Protein Precipitation solution to RNase-treated samples and vortexing vigorously at high speed for 20 s and centrifuging for 3 min at 13,000× *g*. The precipitated proteins formed a tight white pellet and the supernatant containing the DNA was transferred carefully to a clean 1.5 mL-tube containing 600 μL of room temperature isopropanol (Sigma-Aldrich, St. Louis, MO, USA) and the solution was gently mixed by inversion until white threadlike strands of DNA formed a visible mass and then the solution was centrifuged for 1 min at 13,000× *g* at room temperature. Then, using 600 μL 70% ethanol (Sigma-Aldrich, St. Louis, MO, USA) and interval several times to wash DNA. Finally, the DNA was rehydrated in the DNA rehydration solution by incubating at 65 °C for 1 h and the genomic DNA concentration was subsequently determined at OD 260/280 and OD 260/230. The genomic DNA was stored at 4 °C.

### 2.7. PCR

Expression of the adenoviral E1A gene was confirmed by PCR using forward E1A primer and as internal control (primer sequence are list on Appendix A). The amplification was carried out in a total volume of 25 μL containing 500 ng genomic DNA, 0.4 μM of each primer, 200 μM dNTP (Takara Biochemicals, Otsu, Japan) 1.5 mM MgCl_2_ (Roche, Basel, Switzerland), 1U FastStart Taq polymerase (Roche, Basel, Switzerland), and 1× PCR reaction buffer (Roche, Basel, Switzerland). The entire mixture of 25 μL was subjected to 35 cycles of 1 min denaturation at 95 °C, 1 min to allow annealing at 63 °C, and 2 min of extension at 72 °C. During the last cycle, the extension time was increased by 7 min. During the last cycle, the extension time was increased by 7 min. Amplified products were analyzed by 1.5% agarose gel electrophoresis.

### 2.8. Quantitative Real-Time PCR

The method of quantitative real-time RT-PCR was performed as described [26]. Total RNA (2 μg) of each sample reversely transcribed in 20 μL using 0.5 μg of oligo (dT) (Invitrogen, Carlsbad, CA, USA) and 200U Superscript III RT (Invitrogen, Carlsbad, CA, USA). Amplification was carried out in a total volume of 20 μL, with SYBR Green PCR MasterMix (Applied Biosystems, Foster City, CA, USA), the cDNA and 500 nM of each primer. All of Primer sequences are list on Appendix A. The reaction conditions were one cycle at 95 °C for 10 min followed by 40 cycles of denaturation at 95 °C for 15 s, annealing at 56 °C for 15 s, and extension at 72 °C for 40 sec. Standard curves (cycle threshold values versus template concentration) were prepared for each target gene and for the endogenous reference (GAPDH) in each sample. The quantification of the unknown samples was performed by the ABI Applied Biosystems (Foster City, CA, USA) with StepOne software v2.0 (Applied Biosystems, Darmstadt, Germany).

### 2.9. Human Xenograft HT29 Colorectal Cancer Model

All experimental procedures involving animals were reviewed, approved, and conducted in accordance with established guidelines of the Animal Care and Use Committee at China Medical University. All efforts were also made to minimize animal suffering and to reduce the number of animals used according to the 3Rs principles (replacement, reduction, and refinement). To establish peritoneal colorectal cancer mice models, eight-week-old male athymic Balb/c mice were s.c. injected with 1 × 10^6^ HT29 colon carcinoma cells. After 14 days, the mice were randomly divided into three groups of 9 mice each and IV injected once with 1 × 10^6^ control primed MSCs (MSC), CRAdNTR-loaded primed MSCs (MSC^CRAdNTR^), or 1 × 10^9^ CRAdNTR suspended in the 1:10 diluted NAbs-containing human serum (50 μL serum in 500 μL PBS). Two days later, all mice were i.p. administrated with 25 mg/kg CB1954 (Sigma-Aldrich, St. Louis, MO, USA) for 5 consecutive days per week for 2 cycles. After 14 days of last cycle of CB1954 administration, all mice were given third cycles of 25 mg/kg CB1954. The mice were sacrificed for harvest of tumor and vital organ tissues 7 days later. Tumors from all groups of mice were measured every week with calipers after treatment. The volume was estimated by the formula: volume = width^2^ × length/2.

### 2.10. Immunohistochemistry

Resected xenograft tumor fixed in formalin, paraffin-embedded, and sectioned at 5 μm thickness. Paraffin-embedded xenograft tumor sections were deparaffinized, rehydrated, and antigen retrieved by sodium citrate buffer pH 6.0 at 100 °C for 30 min. Endogenous peroxidase activity in tissue was blocked with 3% hydrogen peroxide (Sigma-Aldrich, St. Louis, MO, USA). Residual enzymatic activity was removed by washing in PBS, and non-specific staining was blocked with Ultra V block (Thermo Scientific, Fremont, CA, USA) for 10 min and incubated with primary antibodies against adenovirus type 5 (1:800, Ad-5,rabbit polyclonal antibody, ab6982, Abcam, Cambridge, MA, USA) and NTR was detected using polyclonal sheep anti-NTR antibody—which was kindly provided by Dr. Chen, MJ, of Mackay Memorial Hospital, Taipei—at 4 °C overnight, washed extensively with PBS. After that, for primary Ad-5 antibody, the sections were incubated at room temperature with secondary polymeric antibody following the manufacturer’s instruction (Super SensitiveTM IHC Detection Systems Kit, BioGenex, Milan, Italy). For primary NTR antibody, reacted with an HRP conjugated secondary rabbit anti-sheep antibody (1:5000, 31480, Thermo Scientific Pierce, USA) at room temperature for 1 h and followed by diaminobenidine (DAB) staining (LSAB kit, Dako, Carpinteria, CA, USA). Finally, counterstaining was performed with Mayer’s hematoxylin (Sigma-Aldrich, St. Louis, MO, USA) and observed with a microscope.

### 2.11. UPLC/MS-QTOF

#### 2.11.1. Sample Preparation

For samples containing FBS, 2 v methanol was added and the samples were held at −20 °C to precipitate proteins. Then, centrifuged at 5000 rpm for 15 min and transferred the supernatant to clear tubes. For tissue sample, to extract the metabolites the 100 mg tissue, including heart, liver, lung, kidney, and xenograft tumor, were ground micropesles followed by adding 500 mL of 50% acetonitrile (ACN, JT-baker, Deventer, Holland), then, homogenized by using microtube homogenizer (BeadBug^TM^, Benchmark, Edison, NJ, USA), the homogenous samples were centrifuged at 13,000 rpm for 10 min, harvested the supernatant and transferred to clear tubes. To remove matrix effect, samples were cleaned-up by off-line solid-phase extraction (SPE) using Oasis HLB™ Cartridge 1 cc (30 mg) (Waters, Wexford, Ireland). The SPE cartridges were conditioned with 1.0 mL methanol followed by 1.0 mL water Milli-Q and the samples were loaded in the cartridge. Then, cartridges were washed with 1 mL of 5% (*v*/*v*) methanol in water Milli-Q. The analytes were eluted by 1 mL absolute methanol. The elution was under a stream of nitrogen. The extracted sample was reconstituted with 100 μL absolute methanol and transferred to an injection vial or store at −20 °C.

#### 2.11.2. Chromatographic System

The chromatographic system consisted of Waters ACQUITY UPLC were equipped with a spectrophotometric with Waters Synapt G1 high-definition mass spectrometer. CB1954 and metabolite separation was performed at 30.0 °C on an analytical ACQUITY UPLC BEH C18 (100 mm × 2.1 mm I.D.) with a particle size of 1.7 µM (Waters). The compounds of interest were analyzed by electrospray ionization-quadrupole time-of-flight mass spectrometry (ESI-QTOF MS). The SYNAPT G1 mass spectrometer was used in V mode and operated in electrospray ionization positive mode with a capillary voltage of 3.0 kV, the sampling cone at 40 V and the extraction cone at 4.0 V. The scan time was 0.2 s covering the range of 50 to 320 Da mass range. The source temperature was 80 °C and the desolvation temperature was set at 250 °C. Nitrogen gas was used as the nebulization gas at a flow rate of 500 L/h. The trap collision energy was set at 6.0 V while the transfer optics collision energy was set to 4.0 V. The MassLynx software (Waters) was used to pilot the ACQUITY UPLC instrument and to process the data (i.e., plotting of chromatograms) throughout the method validation and sample analysis.

#### 2.11.3. Mobile Phase Solutions

The mobile phase is composed of solution A (2% Acetonitrile/H2O + 0.1% Formic acid) and B (100% Acetonitrile + 0.1% Formic acid). Both solutions were degassed by separating with helium. The injection volume was 10 μL. The mobile phase was delivered at 1.0 mL/min. The gradient program conditions are reported in Appendix A.

### 2.12. Pharmacokinetics of CRAdNTR in Venous Blood

An aliquot of 100 μL of 2 × 10^8^ CRAdNTR particles with the 1:10 diluted NAb was infused into the right retro-orbital sinus of three mice. Tail blood samples at 10, 30, 90, 180, and 360 min following the CRAdNTR virus infusion were used for pharmacokinetic studies. The genomes of virus per 50 μL of blood were extracted by DNeasy Blood and Tissue kits (QIAGEN, Germantown, MD, USA) and determined by quantitative-PCR. For quantification, a standard curve was constructed by assaying serial dilutions of CRAdNTR virus ranging from 1 × 10^7^ ~ 1 × 10^3^ vp/mL. The correlation coefficient of the standard curve was 0.9982.

### 2.13. Statistical Analysis

All values are expressed as mean ± SD. Analysis of variance (ANOVA) and Student’s *t*-test were used for statistical comparisons in groups greater than and equal to two, respectively (GraphPad Prism 5.0; GraphPad Software, San Diego, CA, USA). A value of *p* < 0.05 is considered to be statistically significant.

## 3. Results

### 3.1. Trichostatin A (TSA)-Primed MSCs Have a Normal MSC Surface Phenotype

TSA is known to upregulate chemokine receptors, such as CXCR4 [27], and increase CAR expression on bladder cancer cells [28] although its effects on MSCs are unknown. We first demonstrated that TSA at a concentration greater than 150 ng/mL induced cytotoxicity in hypoxic MSCs, as evidenced by crystal violet staining (Appendix A) and the MTT assay (Appendix A). Therefore, we used a concentration of 100 ng/mL to prime MSCs for 24 h. TSA-primed MSCs showed no difference with control MSCs in morphology (Appendix A), proliferation capacity (Appendix A), and differentiation potentials (Appendix A). Moreover, TSA-primed MSCs highly expressed CD29, CD44, and CD105, which are the most commonly reported positive surface markers of MSCs, but did not express the most frequently reported negative surface markers, such as CD31, CD34, and CD45 (Appendix A). More importantly, TSA-primed MSCs did not express a senescence marker (Appendix A). These results together suggest that MSCs primed with TSA have a normal MSC phenotype.

### 3.2. TSA Enhances MSC Tumor Tropism

Since only a small portion of MSCs reach the tumor sites after systemic administration [29,30], optimization of MSC tumor tropism is important for using MSCs to deliver oncolytic virus in the context of various tumor models. CXCR4 is a chemokine receptor responsible for MSC tumor tropism [31]. A previous study showed that histone deacetylase inhibitors (HDACi) increase CXCR4 expression in hematopoietic stem/progenitor cells [32]. We therefore asked whether TSA could increase CXCR4 expression on MSCs. Quantitative RT-PCR and flow cytometry revealed that primed MSCs had significantly upregulated CXCR4 mRNA (Figure 1A) and surface protein levels (Figure 1B). In addition, primed MSCs also showed improved migration ability and cancer cell tropism, e.g., when employing the HT29 human colorectal cancer cell line (Figure 1C,D). These data suggest that TSA treatment can be used to upregulate CXCR4 expression, as well as to enhance tumor tropism in MSCs.

### 3.3. TSA Enhances CAR Expression and Adenovirus Infection Rates

TSA has been reported to increase CAR expression on bladder cancer cells [28]. We then asked whether TSA had a similar effect on MSCs. Both quantitative RT-PCR (Figure 2A) and flow cytometry (Figure 2B) showed that surface expression of CAR was significantly upregulated in MSCs after TSA treatment. When evaluating the viral loading capacity of MSCs via CRAdEGFP infection followed by analysis of EGFP fluorescence intensities, we found that primed MSCs also featured increased viral loading (Figure 2C). These data suggest that TSA treatment increases CAR expression, as well as the susceptibility to adenoviral infection of MSCs.

### 3.4. Combination of MSC-Delivered CRAdNTR with a Prodrug Induces Cytotoxicity in Colorectal Cancer Cells in the Presence of NAb In Vitro

Since preexisting neutralizing antibodies (NAb) in the circulation represent a major obstacle of oncolytic viral therapy upon clinical application, we performed all subsequent experiments in the presence of NAb to mimic conditions reflecting those in clinical practice. To reduce the cytotoxic effects of oncolytic viral therapy on normal host tissues or cells, we investigated the cytotoxic effects on three p53-deficient colorectal cancer cell line, including HT29, SW480, and SW620 [33], through the combination of prodrug activation and MSC-delivered CRAdNTR both in vitro and in vivo. The corresponding experimental procedures are shown in Figure 3A. The data revealed that CRAdNTR alone, blocked by NAb, did not cause cytotoxicity in HT29 cells either with or without CB1954 treatment, suggesting that CRAdNTR did not infect HT29 cells and that CB1954 was not activated. Interestingly, when primed MSCs loaded with CRAdNTR (MSC^CRAdNTR^) were added to HT29 cells, cytotoxicity was only observed in the presence of CB1954, as evidenced by a significant decrease in crystal violet staining (Figure 3B) and the MTT assay (Figure 3C). Notably, co-culture of primed MSCs did not induce any decrease in crystal violet staining or the MTT assay, both in the absence or presence of CB1954, suggesting that primed MSCs alone did not induce cytotoxicity in HT29 cells (Figure 3B,C). Similar results were observed in SW480 and SW620, where significant cytotoxicity was only observed when MSC^CRAdNTR^ were added to these cells in the presence of CB1954 (Appendix A). Furthermore, quantitative RT-PCR for adenoviral E1A confirmed that CRAdNTR, carried by MSC^CRAdNTR^ cells, was delivered into HT29 cells in the presence of NAb (Figure 3D). These results suggest that MSC^CRAdNTR^ successfully protected CRAdNTR from NAb neutralization and could deliver the viruses to colorectal cancer cells. Moreover, the combination of MSC delivery and prodrug CB1954 activation potentially enhanced the in vitro cancer-killing efficacy of CRAdNTR in the presence of NAb.

### 3.5. MSC-Delivered CRAdNTR Combined with a Prodrug Reduces Colorectal Cancer Growth in the Presence of NAb In Vivo

To determine the in vivo cancer-killing effect of CRAdNTR via the combination of MSC delivery and prodrug CB1954 activation in the presence of NAb, the experimental design shown in Figure 4A was employed, where CRAdNTR alone, MSC^CRAdNTR^, and primed MSCs were intravenously injected into tumor-bearing immunodeficient mice. The mice also received CB1954 via i.p. injection every five consecutive days per week at a dosage of 25 mg per kilogram for two cycles (weeks). Tumor growth curves revealed that tumor growth in mice receiving MSC^CRAdNTR^ was significantly abrogated compared to the other two groups that received CRAdNTR or primed MSCs (Figure 4B,C). In addition, to investigate the durability of activated oncolytic adenovirus after inoculation in vivo, CB1954 injection was delayed by about 20 days after the second cycle of CB1954 treatment before the third cycle was initiated. Delayed CB1954 injection also significantly inhibited the increase in tumor volume in the MSC^CRAdNTR^ group at approximately day 50, while the tumors in the other two groups continued to grow. These results suggest that oncolytic adenovirus is a powerful agent that possesses a long-lasting replication activity to induce the cytolytic effect, which might enhance its therapeutic efficacy.

To examine whether oncolytic adenoviruses were successfully transported into solid tumor sites via MSC delivery, xenografted tumors were excised for genomic DNA extraction 9 weeks later, followed by analysis of adenoviral gene expression. As expected, the adenoviral E1A gene was detected in the tumor grafts of mice receiving MSC^CRAdNTR^ (*n* = 5), but not in tumor grafts of mice receiving CRAdNTR or primed MSCs (Figure 4D). In contrast, the viral genome was only detected in plasma within a few hours after intravenous injection and could not be detected 6 h after infusion (Appendix A). Immunohistochemistry further showed that tumors treated with MSC^CRAdNTR^, but not those treated with CRAdNTR or primed MSCs, were positive for NTR and Ad5, the adenoviral capsid protein (Figure 4E). Furthermore, hematoxylin and eosin (H&E) staining revealed a significant increase in necrotic areas in MSC^CRAdNTR^-treated tumor sections compared to those of mice that received CRAdNTR or primed MSCs (Figure 4F), suggesting that the combination of CB1954 activation with MSC-delivered CRAdNTR effectively lysed tumors, and generated areas containing lytic cells. These results suggest that MSCs act as carriers to successfully protect CRAdNTR from Nab-mediated neutralization, and transfer the virus to the tumor site, where it replicates and lyses tumor cells. The combination of the gene suicide system NTR/CB1954 caused cytolytic effects to reduce colorectal cancer growth in vivo.

### 3.6. MSC-Delivered CRAdNTR Combined with a Prodrug Induces Cytotoxicity in Colorectal Cancer Cells In Vitro via the Generation of Cytotoxic Metabolites

*Escherichia coli* NTR has been described to convert CB1954 into cytotoxic metabolites, like 2-hydroxylamine, 4-hydroxylamine, 2-amine, and 4-amine [34]. To confirm this, metabolites present in the extracellular medium of in vitro cultures were deproteinized, concentrated, and analyzed by UPLC-MS/QTOF. The mass spectra gave prominent parent molecular ions at the expected *m*/*z* values: CB1954 ([M + H]^+^ = 253.057), 2-hydroxylamine and 4-hydroxylamine ([M + Na]^+^ = 261.0599), and 2-amine and 4-amine, ([M + Na]^+^ = 245.065). The metabolites of interest were identified by comparing the retention times with those of the standards, which are the products of CB1954 (retention time: 4.7 min) reacted with exogenous *Escherichia coli* NTR enzyme. The metabolites hydroxylamine and amine had retention times of 5.1 and 3.1 min, respectively, as illustrated by the extracted-ion chromatogram in Figure 5. These results indicated that the metabolites from HT29 cells treated with MSC^CRAdNTR^, as well as with CB1954, were detected at retention times of 5.1 and 3.1 min. Furthermore, we could show that the conversion of CB1954 was incomplete. However, metabolites were not found in HT29 cells treated with CRAdNTR or primed MSCs in combination with CB1954. These data suggest that the in vitro cytotoxic effects on colorectal cancer cells were caused by the production of cytotoxic metabolites from the prodrug CB1954.

### 3.7. Cytotoxic Metabolites from CB1954 Specifically Exist in the Tumor Area Rather than in Other Vital Organs

We further demonstrated that toxic metabolites were only detected in tumors of mice receiving MSC^CRAdNTR^, but not in tumors of mice receiving CRAdNTR or primed MSCs in combination with CB1954 treatment (Figure 6A). Notably, the safety issues regarding the use of oncolytic viruses for cancer treatment are well recognized [35]. Although CRAdNTR carried by MSCs was successfully delivered to tumor sites, it was uncertain whether it would also be transferred to vital organs, causing unexpected toxicity. It was revealed that the cytotoxic metabolites, hydroxylamine, and amine, which had single peak profiles, were detectable only in the tumors of MSC^CRAdNTR^-treated mice, but not in other organs, such as the hearts, livers, lungs, and kidneys of these mice (Figure 6A). Metabolites of CB1954 were also not detected in solid tumors and vital organs of mice receiving CRAdNTR and primed MSCs (Figure 6A). Similarly, the adenoviral E1A gene was specifically detected in xenografted tumors, but not in other organs of mice that had received MSC^CRAdNTR^ (Figure 6B), which expressed an internal control, mRPL13a, a murine housekeeping gene (Figure 6C). These results provide evidence in support of the safety of our therapeutic strategy, without causing toxicity to vital organs, since MSC^CRAdNTR^ specifically homed to tumor sites, while the tumor lysis effect was caused by cytotoxic metabolites generated from the prodrug CB1954.

## 4. Discussion

Epigenetic modification with HDACi is a new method to induce certain characteristics of MSCs for clinical applications, which may cause major changes in the transcriptome. A comprehensive transcriptome analysis using high-throughput sequencing of pig bone-marrow derived MSCs, both treated and untreated with TSA for 24 h, revealed that TSA does not affect the expression of surface markers that are currently used to define MSCs [36], such as CD90 (positive marker), CD31, and CD34 (negative markers), and has a higher effect on already expressed genes, while its ability to induce silent gene expression is less [37]. These data suggest MSCs primed with TSA still retain some of the original characteristics of MSCs. TSA affects the expression of genes related to a variety of biological processes, and up-regulates genes involved in development, differentiation, neurogenesis, myogenesis, Wnt signaling pathways [37], and pluripotency [38]. Since current research has not used MSC for regenerative medicine purposes involving proliferation and differentiation potential, the changes in gene expression caused by TSA will not have much impact on its use as a cell carrier for oncolytic viruses.

In the current study, TSA not only increased the CXCR4 level and migration ability of primed MSCs towards cancer cells without affecting MSC viability and phenotype, but also resulted in elevated CAR receptor expression on primed MSCs to promote the adenovirus infection rate, thereby enhancing the capacity of CRAdNTR loading. We further demonstrated that an oncolytic adenovirus combined with a prodrug is a powerful tool in cancer therapy, with the ability to convert CB1954 into potentially cytotoxic metabolites by reacting with the NTR enzyme of CRAdNTR, causing tumor lysis and reduction of colorectal cancer growth, while side effects are minimized by avoiding toxicity for vital organs.

Since NTR/CB1954 holds the potential for use in cancer gene therapy, 4-hydroxylamine is the major cytotoxic metabolite, and 2-amine is the main bystander metabolite with better diffusion ability in cellular multilayers [34]. According to our newly obtained data, metabolites could be successfully generated from CB1954 based on the detection of its *m*/*z* ratio. However, 2-hydroxylamine and 4-hydroxylamine, as well as 2-amine and 4-amine are isomers with the same molecular formulas and weights, but with different chemical structures and functional groups. Therefore, even if we try to distinguish them by tandem mass spectrometry, they cannot be well distinguished (data not shown).

Since these metabolites are very rare, making it difficult to purchase a standard for each metabolite from commercial suppliers, it may not be possible to directly determine which peak belongs to which isomer by simply comparing the retention time with that of the standard. To determine which isoform of each metabolite is produced from CB1954 in colorectal cancer, it will be necessary to elute and determine the retention times of the metabolites by UPLC/MS-QTOF, followed by purification and analysis of their configuration by nuclear magnetic resonance (NMR) spectroscopy.

There are limitations to the current study that warrant discussion. For example, it has not been shown that TSA-primed MSCs increased in vivo tumor tropism in comparison with non-primed control MSCs. This can be improved by in vivo experiment using imaging to demonstrate whether this priming attracts more MSCs to the tumors. However, there is a large amount of evidence showing that increased CXCR4 expression on cell surface can promote tumor tropism in vivo [31,39,40], and efforts were also made in the current study to minimize animal suffering and to reduce the number of animals used. These are the reasons why we did not conduct these animal experiments. However, the transwell migration assay, an alternative in vitro assay, also proved TSA-primed MSCs increased tumor tropism. Together, these data indicate that MSCs primed by TSA not only increase CXCR4 expression, but also enhance tumor tropism in vitro.

As MSCs have been shown to possess the ability to home to metastatic cancers [41], our strategy of combining MSCs carrying oncolytic adenovirus with GDEPT holds potential and provides superiority with regard to cancer gene therapy. Recent studies have demonstrated the potential of oncolytic viruses carrying genes encoding immune-modulating molecules to combine immune checkpoint blockades for cancer therapy [42,43] thus, combination of TSA-primed MSCs with GDEPT is a promising approach for combining immune checkpoint blockades for treatment of metastatic cancers.

## Figures and Tables

**Figure 1 biomedicines-09-00548-f001:**
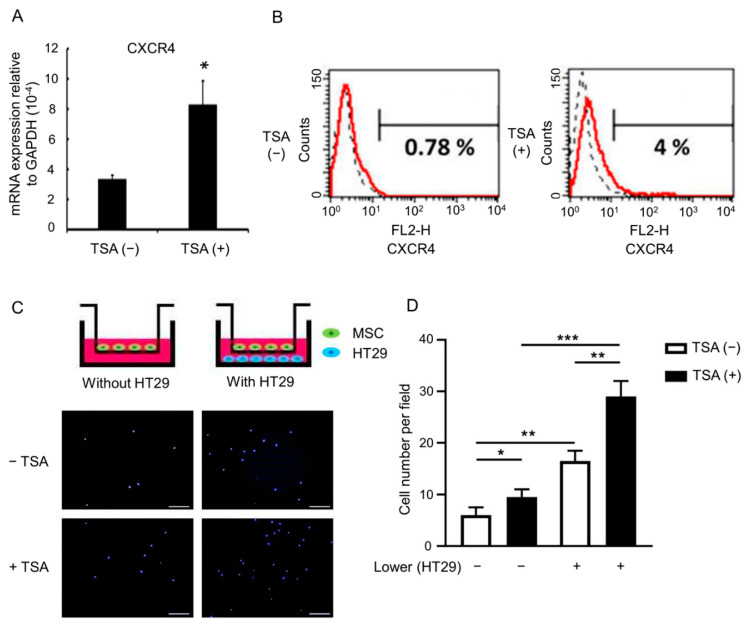
TSA enhances the expression of CXCR4 and the capacity for migration to tumor. MSCs treated without or with 100 ng/mL of TSA treatment for 24 h were analyzed for the expression of CXCR4 using (**A**) quantitative RT-PCR and (**B**) flow cytometry. The results of quantitative RT-PCR are presented as mean ± SD of three independent experiments. (**C**,**D**) In vitro migration abilities of MSCs treated without or with TSA in the absence of presence of HT29 were determined using transwell migration wells. (**C**) The representative photomicrographs showing DAPI-stained cells that migrated to the opposite side of the filter. (**D**) The average numbers of migrating cells per field were assessed by counting four random fields per filter. The quantification results are shown as mean ± SD. Asterisks indicate significant differences as determined by the One-Way ANOVA (* *p* < 0.05, ** *p* < 0.01, *** *p* < 0.001 versus MSCs untreated with TSA). Scale bar, 200 μm.

**Figure 2 biomedicines-09-00548-f002:**
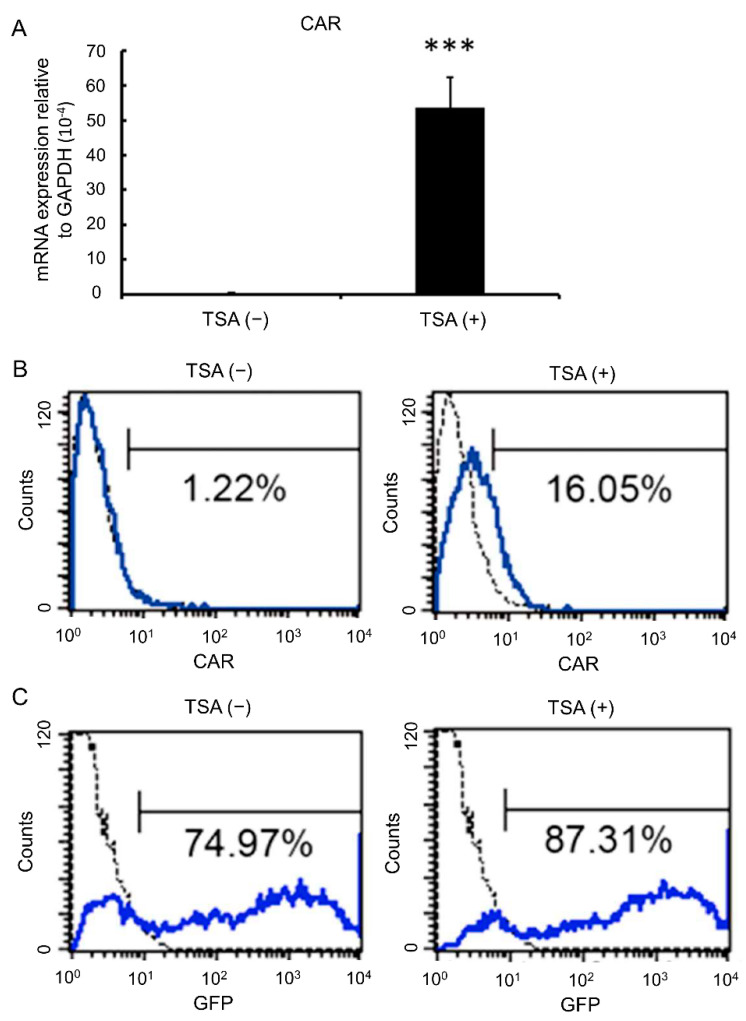
TSA enhances CAR expression and adenoviral infection rate. MSCs treated without or with 100 ng/mL of TSA treatment for 24 h were analyzed for the expression of coxsackievirus and adenovirus receptor (CAR) using (**A**) quantitative RT-PCR and (**B**) flow cytometry. The results of quantitative RT-PCR are presented as mean ± SD of three independent experiments. (**C**) These cells were also subjected to comparison of adenoviral infection rates using CRAdEGFP to infect these cells and analyzing the GFP expression by flow cytometry. Asterisk indicates a significant difference as determined by the Student’s *t* test (*** *p* < 0.001 versus MSCs untreated with TSA).

**Figure 3 biomedicines-09-00548-f003:**
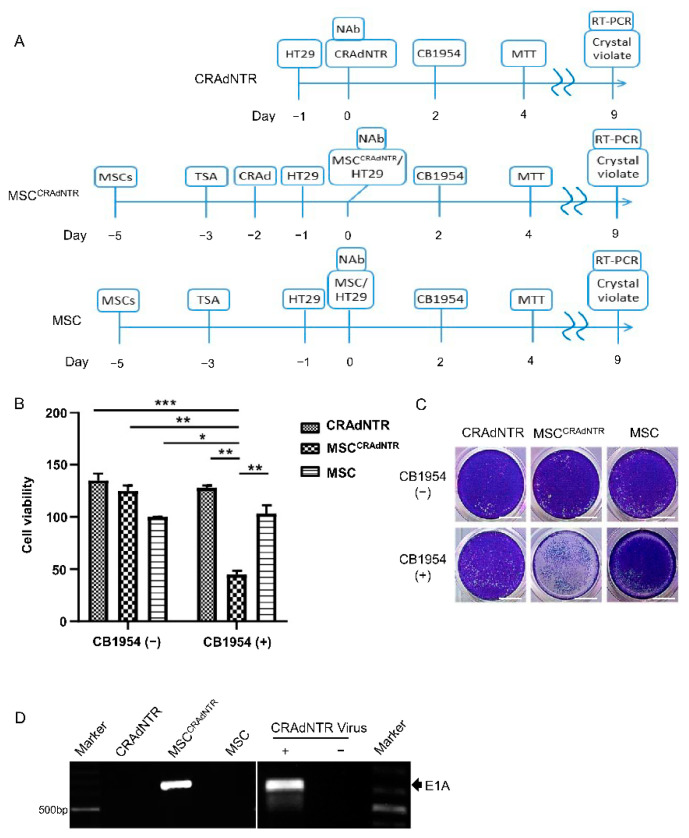
Combination of MSCs-delivered CRAdNTR with prodrug induces cytotoxicity in colorectal cancer cells in the presence of NAb in vitro. (**A**) Flow chart of the in vitro study design. (**B**–**D**) CRAdNTR, MSCs pretreated with 100 ng/mL TSA for 24 h, followed by loading with CRAdNTR (MSC^CRAdNTR^), or MSCs pretreated with TSA alone (MSC) were seeded into wells pre-seeded with HT29. The cell cultures were treated with anti-Adv neutralization antibodies (NAb) and CB1954 at indicated time. Aliquot of cells were subjected to (**B**) MTT assays, (**C**) crystal violet staining, and (**D** Left) RT-PCR detection of the adenoviral E1A gene (675bp) at indicated time. (**D** Right) HT29 treated with and without CRAdNTR virus were used as positive and negative control, respectively. Asterisk indicates a significant difference as determined by One-Way ANOVA (* *p* < 0.05, ** *p* < 0.01, *** *p* < 0.001 versus other groups). Scale bar: 50 mm in (**C**).

**Figure 4 biomedicines-09-00548-f004:**
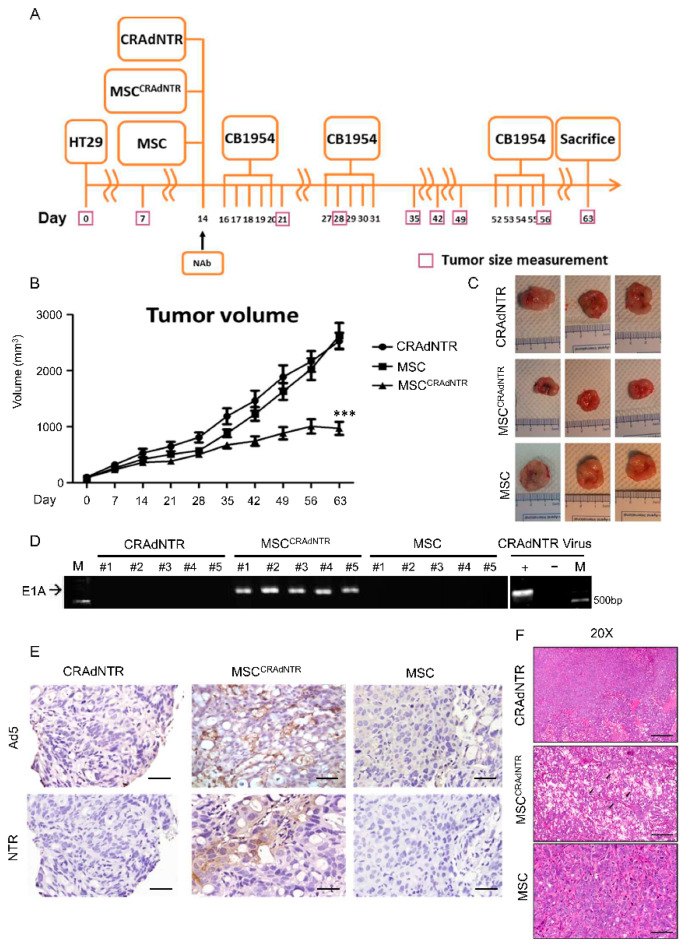
MSCs-delivered CRAdNTR combined with prodrug reduces colorectal cancer growth in the presence of NAb in vivo**.** (**A**) Flow chart of the animal study design. (**B**–**F**) CRAdNTR, MSCs pretreated with 100 ng/mL TSA for 24 h, followed by loading with CRAdNTR (MSC^CRAdNTR^), or MSCs pretreated with TSA alone (MSC) were IV co-infused with anti-Adv neutralization antibodies (NAb) into mice bearing HT29 tumors established 14 days ago. The mice were i.p. injected with CB1954 at indicated times. (**A**) Tumor volumes were measured every week, and the mice were sacrificed and tumors were harvested for analysis. (**C**) Representative macroscopic images of solid tumors. (**D** Left) RT-PCR detection of the adenoviral E1A gene (675 bp), (**D** Right) HT29 treated with and without CRAdNTR virus were used as positive and negative control, respectively. (**E**) IHC detection of Ad5, capsid proteins of adenovirus type 5, and NTR in serial sections, and (**F**) H&E staining for histological analysis. Arrows indicate tumor lysis areas. Asterisk indicates a significant difference as determined by One-Way ANOVA (*** *p* < 0.001 versus other groups). Scale bar: 1 mm (the minimal scale) in (**C**); 50 dμm in (**E**); 100 dμm in (**F**).

**Figure 5 biomedicines-09-00548-f005:**
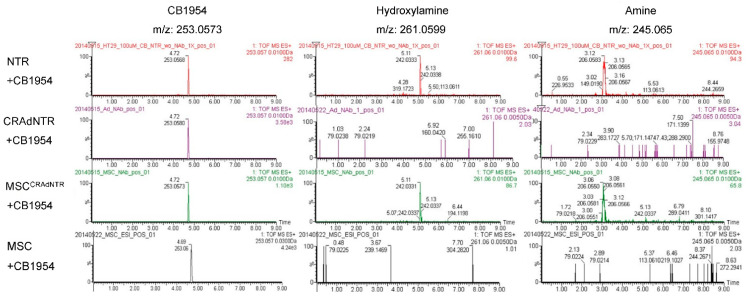
Detection of cytotoxic metabolites of CB1954 in the conditioned medium of HT29 cells. Flow chart of the in vitro study design is the same as Figure 3A. The conditioned medium of HT29 cells treated with CRAdNTR, MSC^CRAdNTR^, or MSC alone, followed by treatment with anti-Adv neutralization antibodies (NAb) and CB1954, were analyzed by UPLC/MS-QTOF. NTR+CB1954 solution referred to as standard metabolites of the reaction of CB1954 with exogenous NTR is showed in upper panel. CB1954 were protonated molecule [M + H]^+^ at *m*/*z* 253.0573. However, hydroxylamine and anime have been ionized by the addition of a sodium cation [M + Na]^+^ at *m*/*z* 261.0599 and 245.0645, respectively. CB1954, hydroxylamine and amine exhibited peaks at retention times 4.7, 5.1, and 3.1 min in the extracted ion chromatogram. The results show that only the conditioned medium from CB1954 treated to MSC^CRAdNTR^ which co-cultured with HT29 cells exhibited peaks of hydroxylamine and amine.

**Figure 6 biomedicines-09-00548-f006:**
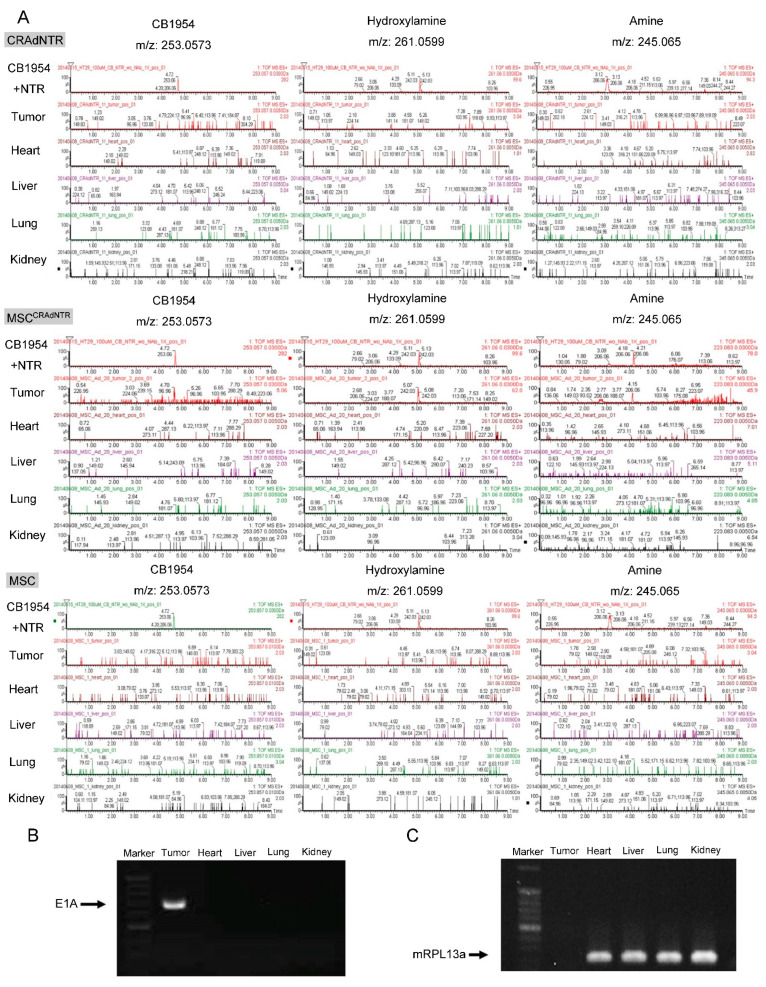
Cytotoxic metabolites of CB1954 were produced only in tumor of MSC^CRAdNTR^ treated mice. Flow chart of the animal study design is the same as Figure 4A. (**A**) Mice bearing HT29 tumors IV infused with CRAdNTR, MSC^CRAdNTR^, or MSC alone, followed by treatment with anti-Adv neutralization antibodies (NAb) and CB1954. The mice were sacrificed for harvest of tumor and vital organ tissues. (**A**) The extracts were analyzed by UPLC/MS-QTOF. NTR+CB1954 solution referred to as standard metabolites of the reaction of CB1954 with exogenous NTR is showed in upper panel. Retention times of CB1954 (4.7 min), hydroxylamine (5.1 min), and amine (3.1 min) are shown in the extracted ion chromatograms of “CB1954+NTR”. RNA was isolated and subjected to RT-PCR analysis for the expression of (**B**) adenoviral E1A gene (675bp) and (**C**) mRPL13a, a mouse housekeeping gene. “M” represented as marker.

## Data Availability

Data is contained within the article.

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
