# Peer review of "Combination of Mesenchymal Stem Cell-Delivered Oncolytic Virus with Prodrug Activation Increases Efficacy and Safety of Colorectal Cancer Therapy"

_biomedicines, 2021, doi:10.3390/biomedicines9050548_

Round 1
Reviewer 1 Report
Dear Authors,
Thank you for your responses. Despite my reservations in relation to your responses on the suggested control experiments, my remaining questions were addressed.
Author Response
Point-by-point responses
Responses to reviewer #1:
Dear Authors,
Thank you for your responses. Despite my reservations in relation to your responses on the suggested control experiments, my remaining questions were addressed.
Response: Thank you very much for your kind opinion.

Reviewer 2 Report
In this manuscript by Ho et al., the authors have studied the efficacy of mesenchymal stem cell delivered oncolytic viruses in combination with prodrug activation to kill colorectal cancer cells both in vitro and in vivo. They concluded that the delivery of oncolytic viruses by MSCs combined with the activation of prodrugs is effective against colorectal cancer cells. This is an interesting study with clinical significance. The following comments should be addressed before reconsidering the manuscript.
- The novelty of the study should be clearly mentioned in the introduction section of the manuscript.
- Pharmacokinetic profile should be provided for adenovirus genome copy number by QPCR.
- Biodistribution of the three MSCs types injected into tumor-bearing immunodeficient mice should be analyzed in different organs.
- Positive control for E1A gene should be included in Figures 3D and 4D.
Author Response
Point-by-point responses
Responses to reviewer #2:
In this manuscript by Ho et al., the authors have studied the efficacy of mesenchymal stem cell delivered oncolytic viruses in combination with prodrug activation to kill colorectal cancer cells both in vitro and in vivo. They concluded that the delivery of oncolytic viruses by MSCs combined with the activation of prodrugs is effective against colorectal cancer cells. This is an interesting study with clinical significance. The following comments should be addressed before reconsidering the manuscript.
- The novelty of the study should be clearly mentioned in the introduction section of the manuscript.
Response: Thank you very much for your kind comments. We try to point out the novelty of this study in the introduction part of the manuscript, below.
“According to these evidences, the combination of CRAd and CB1954 is a new cancer treatment strategy for p53-deficient cancers. However, it remains to be elucidated whether the use of MSC as a cell carrier can further improve its anti-cancer efficacy and safety. In the current study, we first improved the susceptibility of MSCs to CRAd infection, and further increased MSC tumor tropism. We then combined MSC-delivered CRAd and prodrug activation to kill p53-deficient colorectal cancer cells both in vitro and in vivo.” Please refer to Page 2 line 40-46.
- Pharmacokinetic profile should be provided for adenovirus genome copy number by QPCR.
Response: Thank you very much for your great comments. According to your comments, we have now addressed a background regarding this in the “Introduction” section as below.
“The safety and viral pharmacokinetics of CRAd for cancer therapy had been studied in early phase clinical trials [1] and the viral genomes in plasma were only detected in early hours following i.v. infusion with 2´1011 to 2´1012 particles and were not detected at 8 hours [2,3].” Please refer to Page 2 line 10-13.
We have also analyzed the viral pharmacokinetic profile of CRAdNTR following i.v. infusion in mice and the data are included in Figure S4. The method has been added in Page 7 line 4-10 and the result has been added in Page 13 line 18-20.
- Biodistribution of the three MSCs types injected into tumor-bearing immunodeficient mice should be analyzed in different organs.
Response: Thank you very much for your great comments. In previous study [4], we have analyzed biodistribution and performed cell fate tracking of human MSCs following i.v. transfusion into mice bearing human tumor xenografts using lentiviral transduction with herpes simplex virus type 1-thymidine kinase (HSV1-TK) and EGFP gene. In vivo PET images of tumors growing for 4 weeks showed the presence of HSV1-TK+ tumor stroma with an average of 0.36 F 0.24% ID/g [18F]-FHBG accumulation. The levels of micro-PET detectable [18F]-FHBG radioactivity concentration in tumors increased exponentially between days 17 and 30 and paralleled increasing numbers of EGFP+ cells in the growing tumors, suggesting the transplanted human MSCs grew along with tumor growth during this period. We also observed that EGFP+ progeny cells differentiated into tumor stroma, endothelial cells of tumor capillaries and small blood vessels, hair follicle cells, and basal cells of dermis overlaying the s.c. tumor xenografts. We also speculated that the later observations may be explained, at least in part, by the capacity of stem cells to nonspecifically home into the sites of tissue damage and inflammation [5], which are often observed in the skin overlaying s.c. tumors. In the current research, the biodistribution and cell fate tracking of i.v. transfused human MSCs have been studied before, so we did not use any tags to label human MSCs. According to your suggestion, we also address this background in the “Introduction” section as below.
“The biodistribution and cell-fate tracking of human MSCs following i.v. infusion into mice bearing human tumor xenografts have been well studied before [4]. They were mainly detected at the tumor inoculation sites, and grew as the tumors grew, and differentiated into tumor stroma, endothelial cells of tumor capillaries and small blood vessels, hair follicle cells, and basal cells of dermis overlaying the s.c. tumor xenografts [4].” Please refer to Page 2 line 16-21.
- Positive control for E1A gene should be included in Figures 3D and 4D.
Response: Thank you very much for your kind comments. We have now added the positive control for E1A gene in Figures 3D and 4D.
References:
- Huebner, R.J.; Rowe, W.P.; Schatten, W.E.; Smith, R.R.; Thomas, L.B. Studies on the use of viruses in the treatment of carcinoma of the cervix. Cancer 1956, 9, 1211-1218, doi:10.1002/1097-0142(195611/12)9:6<1211::aid-cncr2820090624>3.0.co;2-7.
- Reid, T.; Galanis, E.; Abbruzzese, J.; Sze, D.; Wein, L.M.; Andrews, J.; Randlev, B.; Heise, C.; Uprichard, M.; Hatfield, M.; et al. Hepatic arterial infusion of a replication-selective oncolytic adenovirus (dl1520): phase II viral, immunologic, and clinical endpoints. Cancer Res 2002, 62, 6070-6079.
- Nemunaitis, J.; Cunningham, C.; Buchanan, A.; Blackburn, A.; Edelman, G.; Maples, P.; Netto, G.; Tong, A.; Randlev, B.; Olson, S.; et al. Intravenous infusion of a replication-selective adenovirus (ONYX-015) in cancer patients: safety, feasibility and biological activity. Gene Ther 2001, 8, 746-759, doi:10.1038/sj.gt.3301424.
- Hung, S.C.; Deng, W.P.; Yang, W.K.; Liu, R.S.; Lee, C.C.; Su, T.C.; Lin, R.J.; Yang, D.M.; Chang, C.W.; Chen, W.H.; et al. Mesenchymal stem cell targeting of microscopic tumors and tumor stroma development monitored by noninvasive in vivo positron emission tomography imaging. Clin Cancer Res 2005, 11, 7749-7756, doi:10.1158/1078-0432.CCR-05-0876.
- Cottler-Fox, M.H.; Lapidot, T.; Petit, I.; Kollet, O.; DiPersio, J.F.; Link, D.; Devine, S. Stem cell mobilization. Hematology Am Soc Hematol Educ Program 2003, 419-437, doi:10.1182/asheducation-2003.1.419.

Round 2
Reviewer 2 Report
The authors have addressed all the comments and the manuscript can be accepted for publication.
This manuscript is a resubmission of an earlier submission. The following is a list of the peer review reports and author responses from that submission.
Round 1
Reviewer 1 Report
I reviewed previously this manuscript and my comments are noted below. I have read the resubmitted manuscript and the main point that I was making has not been addressed. This is in my view important as the authors claim that trichostatin A in hypoxia can prime mesenchymal stem cells to enhance their tropism to tumors but they do not demonstrate it in vivo. There is no scientific reason that prevents performing this experiment. As a result, this manuscript is in its current state not acceptable for publication.
The manuscript « Combination of mesenchymal stem cell-derived oncolytic virus with prodrug activation increases efficacy and safety of colorectal cancer therapy” by Chun-Te Ho et al. describes the utilization of mesenchymal stem cells (MSCs) to deliver in vivo an oncolytic adenovirus armed with the E. coli nitroreductase (NTR) gene. The authors show that this approach leads to a therapeutic effect in vivo, even in the presence of adenovirus-neutralizing antibodies.
Although this approach is not entirely new, the paper has merit. In particular, the fact that the MSCs are primed by a treatment of trichostatin A in hypoxia is interesting and novel. This “priming” leads to an increased infectivity of the MSCs to adenovirus but also increases the expression of CXCR4, a factor described to promote migration of the MSCs to the tumors. In my view, the demonstration of the latter point is not complete. The authors demonstrate such increased tropism in vitro but not in vivo. For example, they do not show a “non-primed” control MSCs loaded with adenovirus to evaluate whether the priming is really efficient. Another in vivo experiment required would be to use imaging to demonstrate whether this priming attracts more MSCs to the tumors. In the current version of the paper, these important controls are missing and at least one of these experiments must be performed to validate the approach.
Author Response
Point-by-point responses
Reviewer #1
I reviewed previously this manuscript and my comments are noted below. I have read the resubmitted manuscript and the main point that I was making has not been addressed. This is in my view important as the authors claim that trichostatin A in hypoxia can prime mesenchymal stem cells to enhance their tropism to tumors but they do not demonstrate it in vivo. There is no scientific reason that prevents performing this experiment. As a result, this manuscript is in its current state not acceptable for publication.
The manuscript « Combination of mesenchymal stem cell-derived oncolytic virus with prodrug activation increases efficacy and safety of colorectal cancer therapy” by Chun-Te Ho et al. describes the utilization of mesenchymal stem cells (MSCs) to deliver in vivo an oncolytic adenovirus armed with the E. coli nitroreductase (NTR) gene. The authors show that this approach leads to a therapeutic effect in vivo, even in the presence of adenovirus-neutralizing antibodies.
Although this approach is not entirely new, the paper has merit. In particular, the fact that the MSCs are primed by a treatment of trichostatin A in hypoxia is interesting and novel. This “priming” leads to an increased infectivity of the MSCs to adenovirus but also increases the expression of CXCR4, a factor described to promote migration of the MSCs to the tumors. In my view, the demonstration of the latter point is not complete. The authors demonstrate such increased tropism in vitro but not in vivo. For example, they do not show a “non-primed” control MSCs loaded with adenovirus to evaluate whether the priming is really efficient. Another in vivo experiment required would be to use imaging to demonstrate whether this priming attracts more MSCs to the tumors. In the current version of the paper, these important controls are missing and at least one of these experiments must be performed to validate the approach.
Response: Thank you very much for your great comments. We also agree that there is no in vivo evidence that TSA-primed MSC has an increased tumor tropism, but due to time constraints and other reasons, we cannot conduct this study. Instead, we have now added a discussion in the Discussion section. “There are limitations to the current study that warrant discussion. For example, it has not been shown that TSA-primed MSCs increased in vivo tumor tropism in comparison with non-primed control MSCs. This can be improved by in vivo experiment using imaging to demonstrate whether this priming attracts more MSCs to the tumors. However, there are a large amount of evidence showing that CXCR4 can promote tumor tropism in vivo [1-3], and efforts were also made in the current study to minimize animal suffering and to reduce the number of animals used according to the 3Rs principles (replacement, reduction, and refinement). These are the reasons why we did not conduct these animal experiments. However, the transwell migration assay, an alternative in vitro assay, also proved TSA-primed MSCs increased tumor tropism. Together, these data indicate that MSCs primed by TSA not only increase CXCR4 expression, but also enhance tumor tropism in vitro.” Please refer to Page 16 line 478-486.
Reference:
- Lourenco S, Teixeira VH, Kalber T, Jose RJ, Floto RA, Janes SM. Macrophage migration inhibitory factor-CXCR4 is the dominant chemotactic axis in human mesenchymal stem cell recruitment to tumors. J Immunol. 2015; 194: 3463-74.
- Kalimuthu S, Oh JM, Gangadaran P, Zhu L, Lee HW, Rajendran RL, et al. In Vivo Tracking of Chemokine Receptor CXCR4-Engineered Mesenchymal Stem Cell Migration by Optical Molecular Imaging. Stem Cells Int. 2017; 2017: 8085637.
- Park SA, Ryu CH, Kim SM, Lim JY, Park SI, Jeong CH, et al. CXCR4-transfected human umbilical cord blood-derived mesenchymal stem cells exhibit enhanced migratory capacity toward gliomas. Int J Oncol. 2011; 38: 97-103.

Reviewer 2 Report
Dear Sir/Madam,
Thank you for the opportunity to review the above manuscript submitted to the journal Biomedicines.
The manuscript investigates the use of bone marrow-derived stem cells as vectors for oncolytic viral delivery to tumor tissues. First, the Authors investigated measures to increase tumor tropism and adenovirus susceptibility of BMCSs. For the former one, upregulation of the chemokine receptor CXCR4 was measured. For the latter one, increased expression of the viral infection mediating coxsackievirus and adenovirus receptor was tested. For gene inductions, the Authors used the histone deacetylase inhibitor Trichostatin A. Their conclusion was that Trichostatin A improved both tumor tropism and adenovirus susceptibility of BMSCs. Then, oncolytic Ad-infected BMSCs were tested as viral vectors to target cancer cells both in vivo and in vitro. The Authors’ conclusion was that Trichostatin A-pretreated Ad-infected BMSCs successfully targeted cancer cells and delivered recombinant viral genes to provoke the formation of apoptotic metabolites.
The idea of the use of viruses against tumors has been century-long by now but one of the practical hurdles is the recipient’s immune reaction to oncolytic virus particles. Thus, the idea of the use of immunologically privileged BMSCs is both innovative and timely. The combination of this strategy with the use of suicide genes, that is in the frontline of anticancer research, likely to attract the attention of the interested scientific community and as such, worth to be considered for publication. The current version of the manuscript, however, raises a number of criticisms detailed below.
Major criticism:
- Induction of CXCR4 and CAR was achieved by the use of a histone deacetylase inhibitor. This treatment, however, likely has global effects on chromatin structure that might lead to significant change of the transcriptome. To ensure cells could still be considered stem cells after the histone deacetylase inhibitor treatment and the safety of the use histone deacetylase inhibitors on BMSCs, analyses of the global effect of histone deacetylation on the gene expression pattern seems to be inevitable. Still, the Authors studied the effect of histone deacetylation inhibition on the stemness of treated cells by measuring the expression of few common stem cell markers only but not gene expression at the level of the transcriptome. Alternatively, Authors should demonstrate that characteristics of the histone deacetylase inhibitor-treated BMSCs remain comparable to that of the untreated cells in the context of proliferation, differentiation and senescence using standardized assays.
- Figures need to be reconsidered to provide maximum information to the audience. Accordingly, in Figure 1A, it is not clear what 10 to the power of 4 means on the Y axis if this is relative expression as per axis legend. If it meant fold increase, authors need to show what represents 100%. Quantification of Figure 1B is missing. Figure 1C‘s quality should be improved; the most significant image is the darkest making its evaluation difficult. In Figure 4D, it is not clear what significance indicate. In this experiment, cross correlation would be interesting e.g. TSA+ cells vs. TSA+ cells in both the presence and absence of HT29 cells. As such, however, ANOVA is needed instead of Student’s t-test, so the figure should be recalculated and redrawn.
- In Figure 2A, it is not clear what 10 to the power of 4 means on the Y axis if this is relative expression as per axis legend. If it meant fold increase, authors need to show what represents 100%.
- Based on Figure 2B and C data, BMSCs showed 10-fold increase in CAR expression post TSA treatment. This, however, did not correlate with viral GFP expression of TSA-treated Ad-infected cells. Moreover, the proportion of GFP-positive cells were quite high in TSA-negative cells raising the question if CAR is the primary entry site of Ad in BMSCs. It would be great to see if the expression of viral genes shows similar pattern than that of the GFP.
- In the in vitro co-culture experiment, some controls e.g. CRAdNTR alone is missing. More importantly, the origin of the detected Adenoviral E1A cannot be determined in the experimental setting applied. I was wondering if it would be useful to see the spatio-temporal distribution of viral genes in co-cultures to distinguish if toxic metabolites are produced by transduced cancer cells or the carrier BMSCs.
Minor criticism:
English editing is needed including:
- Text needs to be rephrased at certain parts including lines 95-96
- Typos need to be corrected in lines 62, 127, 131
- Text needs editing in lines 291
Author Response
Point-by-point responses
Reviewer #2
The manuscript investigates the use of bone marrow-derived stem cells as vectors for oncolytic viral delivery to tumor tissues. First, the Authors investigated measures to increase tumor tropism and adenovirus susceptibility of BMCSs. For the former one, upregulation of the chemokine receptor CXCR4 was measured. For the latter one, increased expression of the viral infection mediating coxsackievirus and adenovirus receptor was tested. For gene inductions, the Authors used the histone deacetylase inhibitor Trichostatin A. Their conclusion was that Trichostatin A improved both tumor tropism and adenovirus susceptibility of BMSCs. Then, oncolytic Ad-infected BMSCs were tested as viral vectors to target cancer cells both in vivo and in vitro. The Authors’ conclusion was that Trichostatin A-pretreated Ad-infected BMSCs successfully targeted cancer cells and delivered recombinant viral genes to provoke the formation of apoptotic metabolites.
The idea of the use of viruses against tumors has been century-long by now but one of the practical hurdles is the recipient’s immune reaction to oncolytic virus particles. Thus, the idea of the use of immunologically privileged BMSCs is both innovative and timely. The combination of this strategy with the use of suicide genes, that is in the frontline of anticancer research, likely to attract the attention of the interested scientific community and as such, worth to be considered for publication. The current version of the manuscript, however, raises a number of criticisms detailed below.
Response: Thank you very much. Now, we have responded to each criticism below.
Major criticism:
- Induction of CXCR4 and CAR was achieved by the use of a histone deacetylase inhibitor. This treatment, however, likely has global effects on chromatin structure that might lead to significant change of the transcriptome. To ensure cells could still be considered stem cells after the histone deacetylase inhibitor treatment and the safety of the use histone deacetylase inhibitors on BMSCs, analyses of the global effect of histone deacetylation on the gene expression pattern seems to be inevitable. Still, the Authors studied the effect of histone deacetylation inhibition on the stemness of treated cells by measuring the expression of few common stem cell markers only but not gene expression at the level of the transcriptome. Alternatively, Authors should demonstrate that characteristics of the histone deacetylase inhibitor-treated BMSCs remain comparable to that of the untreated cells in the context of proliferation, differentiation and senescence using standardized assays.
Response: Thank you very much for your suggestion. The characteristics of the histone deacetylase (HDAC) inhibitor-treated BMSCs have been a hot issue in recent years. Because the due date of revisions is 10 days and this study focus on the combination of oncolytic viruses delivered by TSA-primed MSCs with the activation of prodrugs is a new treatment strategy in p53-deficient colorectal tumors, the characteristics of BMSCs isn’t mainly subject in this study. We try to add your suggestion to the discussion part in manuscript, below.
“Epigenetic modification with HDACi is a new method to induce certain characteristics of MSCs for clinical applications, which may cause major changes in the transcriptome. A comprehensive transcriptome analysis using high-throughput sequencing of pig bone-marrow derived MSCs, both treated and untreated with TSA for 24 hours, revealed that TSA does not affect the expression of surface markers that are currently used to define MSCs [1], such as CD90 (positive marker), CD31 and CD34 (negative markers), and has a higher effect on already expressed genes, while its ability to induce silent gene expression is less [2]. These data suggest MSCs primed with TSA still retain some of the original characteristics of MSCs. TSA affects the expression of genes related to a variety of biological processes, and up-regulates genes involved in development, differentiation, neurogenesis, myogenesis, Wnt signaling pathways [2], and pluripotency [3]. Since current research has not used MSC for regenerative medicine purposes involving proliferation and differentiation potential, the changes in gene expression caused by TSA will not have much impact on its use as a cell carrier for oncolytic viruses.”
- Figures need to be reconsidered to provide maximum information to the audience. Accordingly, in Figure 1A, it is not clear what 10 to the power of 4 means on the Y axis if this is relative expression as per axis legend. If it meant fold increase, authors need to show what represents 100%.
Response: Thank you very much for your kind comment. We apologize for the lack of clarity in the original version. The label in the Y axis should be “mRNA expression relative to GAPDH, the housekeeping gene. The label of Figure 1A has now been modified in the revised version.
Quantification of Figure 1B is missing.
Quantification of Figure 1B has also been added in the revised version.
Figure 1C‘s quality should be improved; the most significant image is the darkest making its evaluation difficult.
We have also improved the quality of Figure 1C in the revised version. The new figures are more clear to indicate the significant increase in transwell migration assay.
In Figure 4D, it is not clear what significance indicate.
The figures 4D to 4E were corrected in Page 11 line 365-375. “As expected, the adenoviral E1A gene was detected the tumor grafts of mice receiving MSCsCRAdNTR (n = 5), but not in tumor grafts of mice receiving CRAdNTR or primed MSCs (Figure 4D). Immunohistochemistry further showed that tumors treated with MSCsCRAdNTR, but not those treated with CRAdNTR or primed MSCs, were positive for NTR and Ad5, the adenoviral capsid protein (Figure 4E). Furthermore, hematoxylin and eosin (H&E) staining revealed a significant increase in necrotic areas in MSCCRAdNTR-treated tumor sections compared to those of mice that received CRAdNTR or primed MSCs (Figure 4F), suggesting that the combination of CB1954 activation with MSC-delivered CRAdNTR effectively lysed tumors, and generated areas containing lytic cells. These results suggest that MSCs act as carriers to successfully protect CRAdNTR from NAb-mediated neutralization, and transfer the virus to the tumor site, where it replicates and lyses tumor cells.”
In this experiment, cross correlation would be interesting e.g. TSA+ cells vs. TSA+ cells in both the presence and absence of HT29 cells. As such, however, ANOVA is needed instead of Student’s t-test, so the figure should be recalculated and redrawn.
About the statistic issue, the data with multiple groups (eg. Figure 3B) was analyzed by ANOVA and the data with two groups was analyzed by Student’s t-test. We have also recalculated and redrawn the significance of Figure 1C by one way ANOVA.
- In Figure 2A, it is not clear what 10 to the power of 4 means on the Y axis if this is relative expression as per axis legend. If it meant fold increase, authors need to show what represents 100%.
Response: Thank you very much for your kind comment. We apologize for the lack of clarity in the original version. The label in the Y axis should be “mRNA expression relative to GAPDH, the housekeeping gene. The label of Figure 2A has now been modified in the revised version.
- Based on Figure 2B and C data, BMSCs showed 10-fold increase in CAR expression post TSA treatment. This, however, did not correlate with viral GFP expression of TSA-treated Ad-infected cells. Moreover, the proportion of GFP-positive cells were quite high in TSA-negative cells raising the question if CAR is the primary entry site of Ad in BMSCs. It would be great to see if the expression of viral genes shows similar pattern than that of the GFP.
Response: Thank you very much for your kind comment. According to previous reports, CAR is one of the primary entry sites of Adenovirus [4] and blocking CAR by specific anti-CAR monoclonal antibody (RmcB) reduces the Adenoviral transduction and transgene expression in MSCs [5]. Because adenovirus may target MSCs through multiple adenovirus receptors [4], the baseline of GFP-positive percentage in TSA-non-treated MSCs was high. However, we did show in Figure 2B and C, that TSA-treated MSCs not only increased the expression of CAR receptor but also increased the percentage of GFP-positive cells.
- In the in vitroco-culture experiment, some controls e.g. CRAdNTR alone is missing.
Response: Thank you very much for your kind comments. Actually, the CRAdNTR alone control group has already been included in these experiments, that is CRAdNTR (+), CB1954 (-) in Figure 3B and 3C.
More importantly, the origin of the detected Adenoviral E1A cannot be determined in the experimental setting applied. I was wondering if it would be useful to see the spatio-temporal distribution of viral genes in co-cultures to distinguish if toxic metabolites are produced by transduced cancer cells or the carrier BMSCs.
In the current study, we have already demonstrated in Figure 4D that TSA-treated Ad-infected cells increased the adenoviral E1A gene in vivo. Therefore, we did not further study the similarity of viral gene expression in vitro.
Regarding the origin of the detected Adenoviral E1A, we believe that it is derived from HT-29 colorectal cancer cells (p53 deficient), rather than primary MSC cells (p53 normal). The reason is shown below.
Because the adenovirus ONYX-015 that replicates conditionally through the E1B-55k gene deletion is a potential candidate for the treatment of p53-deficient cancers, it replicates the least in cells with a functional p53 pathway [6], which indicates that only HT29 tumors can produce NTR enzyme. The prodrug, CB1954, is metabolized to 4-hydroxylamine and 2-amine by NTR and suppresses tumor growth. Compared to the in vivo animal experiment, in Figure 5 and 6A, only the HT29 tumor site but not other organs of MSCCRAdNTR group presented toxic metabolites. Thus, we believe the detected Adenoviral E1A is derived from HT-29 colorectal cancer cells.
Minor criticism:
English editing is needed including:
- Text needs to be rephrased at certain parts including lines 95-96
Response: Thank you very much for your kind comment. Text in lines 95-96 has been rephrased as “HT29, a p53-mutant human colorectal cancer cell line [25], was obtained from the american type culture collection (ATCC, Rockville, MD) and grown in dulbecco’s modified eagle medium (DMEM, Gibco) supplemented with 10% FBS.”
- Typos need to be corrected in lines 62, 127, 131
Response: Thank you very much for your kind suggestions. Typos in lines 62, 127, 131 have been corrected.
- Text needs editing in lines 291
Response: Thank you very much for your kind suggestions. Text in lines 291 has been edited as “Asterisks indicate significant differences as determined by the One Way ANOVA (*p<0.05, **P<0.01 versus MSCs untreated with TSA). Scale bar, 200 μm.”
Reference:
- Dominici M, Le Blanc K, Mueller I, Slaper-Cortenbach I, Marini F, Krause D, et al. Minimal criteria for defining multipotent mesenchymal stromal cells. The International Society for Cellular Therapy position statement. Cytotherapy. 2006; 8: 315-7.
- Gurgul A, Opiela J, Pawlina K, Szmatola T, Bochenek M, Bugno-Poniewierska M. The effect of histone deacetylase inhibitor trichostatin A on porcine mesenchymal stem cell transcriptome. Biochimie. 2017; 139: 56-73.
- Samiec M, Opiela J, Lipinski D, Romanek J. Trichostatin A-mediated epigenetic transformation of adult bone marrow-derived mesenchymal stem cells biases the in vitro developmental capability, quality, and pluripotency extent of porcine cloned embryos. Biomed Res Int. 2015; 2015: 814686.
- Adenovirus receptors: implications for targeting of viral vectors, Trends Pharmacol Sci . 2012 Aug;33(8):442-8.
- Lineage Differentiation-Associated Loss of Adenoviral Susceptibility and Coxsackie-Adenovirus Receptor Expression in Human Mesenchymal Stem Cells, STEM CELLS 2004;22:1321–1329.
- An adenovirus mutant that replicates selectively in p53-deficient human tumor cells. Science. 1996; 274: 373-6.

Round 2
Reviewer 1 Report
In my view, the excuses provided by the authors are simply not valid and are boarding to ridiculous. It is a simple and rapid experiment and any decent journal should not accept to publish unfinished reports.
Author Response
Point-by-point responses
Reviewer #1
In my view, the excuses provided by the authors are simply not valid and are boarding to ridiculous. It is a simple and rapid experiment and any decent journal should not accept to publish unfinished reports.
Response: Thank you very much for your great comments. We agree with you the importance of the in vivo experiment using imaging to demonstrate whether this priming attracts more MSCs to the tumors. However, there are a large amount of evidence showing that increased CXCR4 expression on cell surface can promote tumor tropism in vivo [1-3] and the alternative in vitro transwell migration assay has already demonstrated the increased CXCR4 mRNA level (Fig. 1A) and surface expression (Fig. 1B) caused by TSA treatment were associated with increased tumor tropism capacity of MSCs (Fig. 1C). To prepare tumor tropism in vivo studies, we need to construct lentiviral vectors (MSCs can only be efficiently transfected using lentiviral transduction) carrying thymidine kinase (TK) or luciferase reporter genes for micro-PET or IVIS bioluminescence analysis of MSC tracking. Moreover, the colorectal cancer cells with GFP transgene should also be prepared. After those preparations, the immunodeficiency mice should be inoculated with tumor cells, followed by tail vein transplantation of MSCs. The timing of MSC transplantation should also be optimized through many experiments. Therefore, in order to conduct these in vivo experiments, I think it will take a lot of time at the moment, which may delay the publication of the information in the current study. For this limitation, we tried to add a discussion in the Discussion section. Please refer to Page 16 line 480-489.
Reference:
- Kalimuthu, S.; Oh, J.M.; Gangadaran, P.; Zhu, L.; Lee, H.W.; Rajendran, R.L.; Baek, S.H.; Jeon, Y.H.; Jeong, S.Y.; Lee, S.W.; et al. In Vivo Tracking of Chemokine Receptor CXCR4-Engineered Mesenchymal Stem Cell Migration by Optical Molecular Imaging. Stem Cells Int 2017, 2017, 8085637, doi:10.1155/2017/8085637.
- Lourenco, S.; Teixeira, V.H.; Kalber, T.; Jose, R.J.; Floto, R.A.; Janes, S.M. Macrophage migration inhibitory factor-CXCR4 is the dominant chemotactic axis in human mesenchymal stem cell recruitment to tumors. J Immunol 2015, 194, 3463-3474, doi:10.4049/jimmunol.1402097.
- Park, S.A.; Ryu, C.H.; Kim, S.M.; Lim, J.Y.; Park, S.I.; Jeong, C.H.; Jun, J.A.; Oh, J.H.; Park, S.H.; Oh, W.; et al. CXCR4-transfected human umbilical cord blood-derived mesenchymal stem cells exhibit enhanced migratory capacity toward gliomas. Int J Oncol 2011, 38, 97-103.
